# Method for Determining the Susceptibility of the Track

Jerzy Kisilowski and Rafał Kowalik *

Department of Avionics and Control Systems, Faculty of Aviation Division, Military University of Aviation, 08-521 Deblin, Poland
* Correspondence: r.kowalik@law.mil.pl

**Abstract:** This paper discusses real-world experiments in which selected ground imperfections were dynamically analysed in terms of track susceptibility for a linear and non-linear system. The imperfections included wheel pressure on local ground irregularities within the railway turnout. In this study, susceptibility was assumed as an element between two points (two masses)—most often susceptibility is assumed as a Voigt model (parallel combination of stiffness and damping). The tests were carried out for two configurations of train passage through a railway turnout on straight and diverging track. The track stiffness parameters of the railway turnout were determined from deflection measurements measured by sensors positioned at different points of the turnout. The components of the railway turnout were loaded with different forces. The damping parameter was determined from bench measurements of the actual track component. The function describing this damping is dry friction and such a function was determined. The second part of the study was concerned with measurements of stiffness and moments of inertia in a railway switchyard. The analysis carried out indicated the significance of the adverse effects of selected factors on the operation of the railway track (e.g., increasing the length of non-contact of the track with the ground or additional deflections of the railway track rail arising). The paper points out that such imperfections, in addition to the calculated additional deflections, cause, among other things, disturbances in the progressive movement of the rolling stock.

**Keywords:** susceptibility; railways; turnout; track

## 1. Introduction

Many studies [1–5] assume a stiffness value without stating how it was determined for straight track traffic without a turnout. In equations of motion describing the dynamics of passing through a turnout, in many studies [6–8] the value of stiffness is assumed to be constant in magnitude as it is on track without a turnout.

There is a stamping in the track bed, the movement of which has the unambiguous character of dry friction. This follows from elementary principles of physics. Such a characterisation of dry friction has not been encountered in any publication. Viscoelastic damping is usually assumed in models describing the dynamics of the turnout passage. Measurements of the magnitude of the damping stiffness in real objects were determined by measuring the deflection of the rail at a number of points caused by the force of the wheelset on the rail. There were eleven such points. There were six types of wheelset loads. The correlation coefficient between load and deflection was then determined for the straight track. Damping was measured on the bench by recording the hysteresis loop for the left and right wheels in the wheelset [9–11].

Deflections were measured using strain gauge transducers with a linearity of plus or minus 15 mm. Such a range made it possible to measure the deflections of the rail when travelling at 5 km/h on a properly maintained track. The results of this test on the actual facility were the stiffness coefficients of the straight track without the turnout, and the stiffness coefficients of the track when passing through the turnout [12–14].

The concept of modernization susceptibility was introduced in [15], which refers to the degree of adaptation of the analyzed railway line section to a change operating speed. Economic and social requirements, aimed at increasing transport and shortening journey times, determine the increase of the speed above the values determined during the design and construction of the operational lines.

Various methods of measuring track stiffness are currently in use. For example, some studies describe the measurement of rail deflection based on camera and laser data, while publication [2] uses a wagon equipped with an additional axle located in the middle of the wagon's length, which is lowered and raised by a hydraulic cylinder. Another method is to measure the difference in deflection of the rail under a light and heavy wagon [5]. Still another approach is to measure the acceleration axles of a wagon (taking into account the contact force) equipped with a system of oscillating masses [3]. Based on the acceleration of the axles, the deflection of the rail is determined. Yet another method is presented in paper [7], where a dynamic inductor with a small mass oscillating at high frequency is used.

Paper [16] presents a method for identifying voids under the sleeper, i.e., areas where the stiffness of the track is reduced, and these areas are treated as damage to the track. This paper also presents a comprehensive analysis of this type of subject in a number of different works. None of these studies deal with the determination of track stiffness, especially in the area of turnouts. Therefore, the authors do not deal with a detailed analysis of this topic. Paper [17] presents issues concerning dynamic analyses of track with hollows under the sleeper. This paper addresses the study of dynamic phenomena in the rail vehicle-track contact for sleeper voids. The studies [18] deal with the study of phenomena when there are places in the track where the sleeper is not supported by ballast. This task concerns emergency conditions. The authors in this paper deal with track conditions where there are no voids. In paper [19], the authors deal with track failures. This is also a track failure condition. This type of study was not dealt with by the authors of this paper. Paper [20] presents theoretical and experimental studies of the ballast consolidation under the vibration loading of the sleeper. The practical laboratory study is given by the 1:2.5 scaled physical model of one sleeper and the corresponding ballast layer box. The measurements of ballast pressure and deformations under the vibration loading in the ballast layer and the photogrammetric recording of the ballast flow are carried out. Damage to the track superstructure at the turnout is more complex than damage to sections of track and occurs at a much faster rate. The degradation process of the turnout is caused by the higher dynamic impacts of rail vehicles at the track discontinuities. In the crossover section, the most dangerous defects are non-adhesion of the needle to the resistor, chipping on the rolling surface of the needle and the resistor. In the crossover section, defects and damage to the turnout necessitating restricted operation include: crushing of the crossover bow, cracking of the crossover, and run-off in the wing rail glides. The turnouts in the cross member section are subjected to uneven dynamic loading, causing transverse cracks at the strongcrete turnouts. In experimental studies, the susceptibility is assumed as an element between two points (two masses)—most often the susceptibility is assumed as a Voigt model (parallel combination of stiffness and damping). The results obtained for stiffness (especially in a railway turnout) in the form of dry friction damping are functions that have not been determined experimentally in any of the known publications. The purpose of this paper is to outline a new method for testing the susceptibility of railway pavements, which is based on the principle of vibration-based damping determination. The method is completely non-destructive, requiring no disassembly of any pavement element.

## 2. Experimental Testing on a Real Object

In these tests, the stiffness of the track and the magnitude of the damping coefficient with dry friction were determined. The experiment requires the preparation of the freight wagon platforms so that the load in the individual wheelsets is known. Two-axle freight wagons were used. The pressures of the individual wheelsets were recorded using a special

load measuring device. In a track without a turnout, the ground coefficient was assumed to be constant.

### 2.1. Vertical Stiffness Measurements of Straight Track

The loads on the individual axles of the wheelsets are shown in Table 1.

**Table 1.** Axle loads of individual wheelsets.

| No. | Gross Axle Load of the Two-Axle Platforms, N | | |
|:---:|:---:|:---:|:---:|
| | Platform | I Axle | II Axle |
| 1 | 400,000 | 208,000 | 192,000 |
| 2 | 217,000 | 83,000 | 134,000 |
| 3 | 347,000 | 150,000 | 197,000 |
| 4 | 366,000 | 155,000 | 211,000 |
| 5 | 124,000 | 62,000 | 62,000 |
| | Gross load capacity of a coal truck with bogies 25 T N | | |
| | coalminer, N | I *Bogie*, N | II *Bogie*, N |
| 6 | 387,000 | 193,500 | 193,500 |

The train, which was moving at 5 km/h, had a structure like the one in Figure 1.

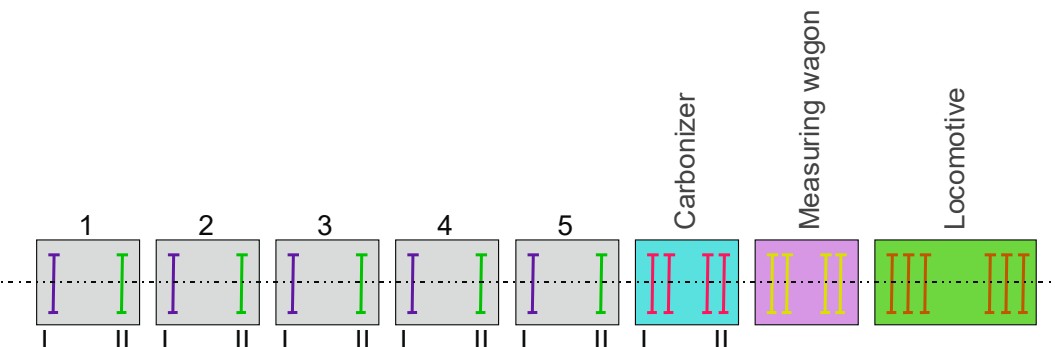

**Figure 1.** Composition of the measuring train.

There are 11 measuring points distributed in the track as shown in Figure 2.

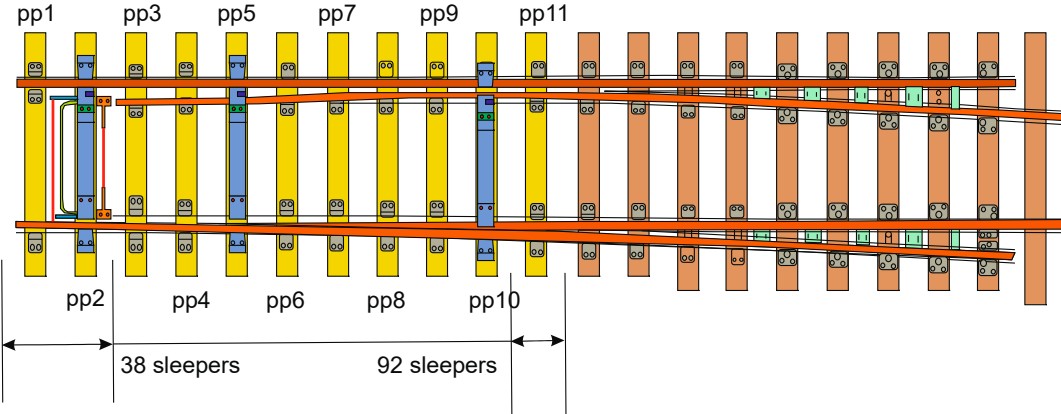

**Figure 2.** Position of measuring points.

The points were positioned in such a way as to eliminate the influence of deflection at neighboring measurement points. The measurement train passing at 5 km/h caused deflection of the track at various measurement points. An example of track deflection is shown in Figure 3.

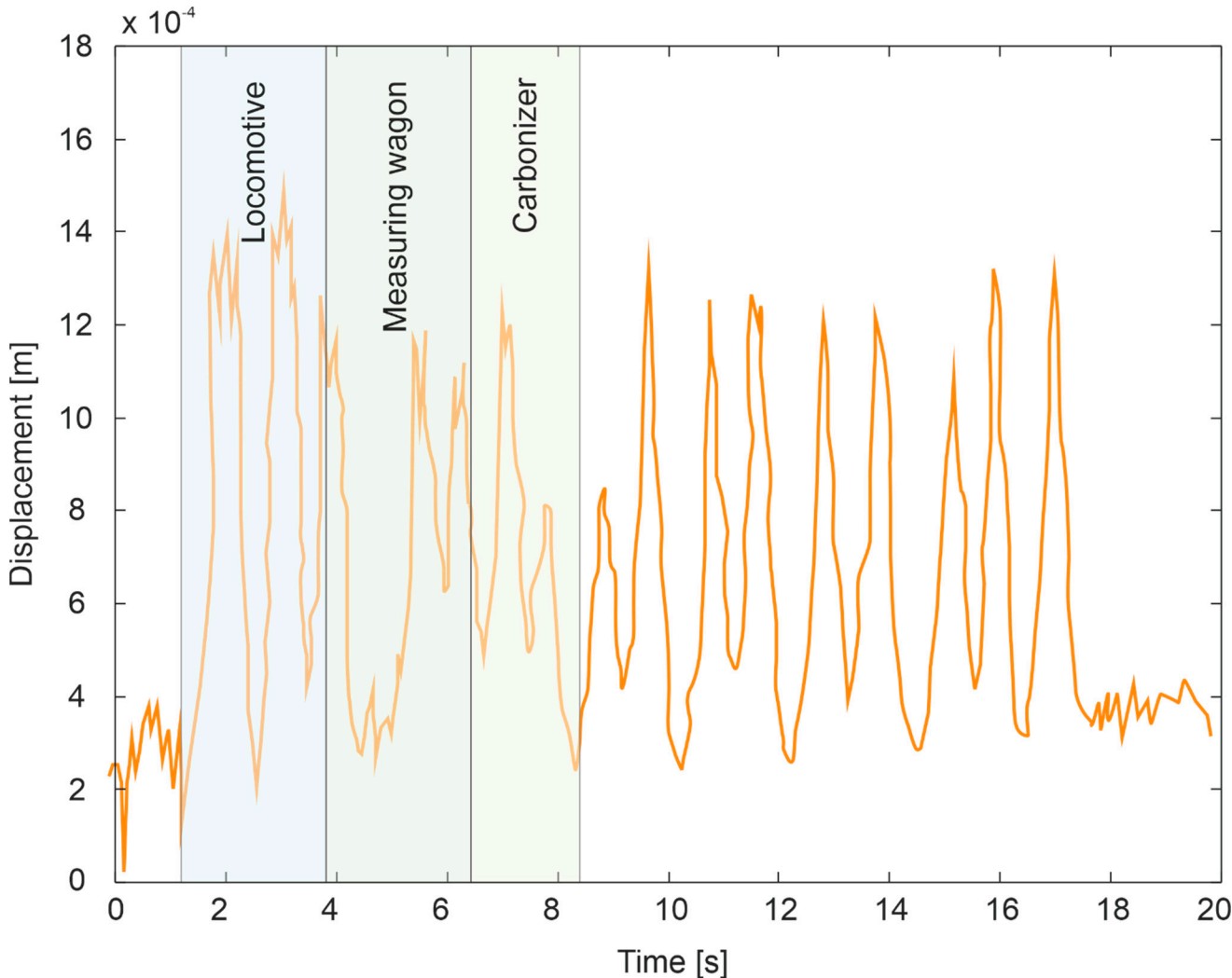

**Figure 3.** Example of track deflection.

Five runs were made, resulting in five deflections at each measurement point. Mean deflection values and standard deviation were determined for these quantities. These results are shown in Table 2.

**Table 2.** Rail deflections.

| $P \cdot 10^4$ N on a Wheel | Mean Values $z \cdot 10^{-4}$ m | | | | | | | | | | |
|---|---|---|---|---|---|---|---|---|---|---|---|
| | p.p.1 | p.p.2 | p.p.3 | p.p.4 | p.p.5 | p.p.6 | p.p.7 | p.p.8 | p.p.9 | p.p.10 | p.p.11 |
| 3.1 | 8.86 | 5.88 | 5.24 | 5.60 | 3.90 | 5.22 | 6.04 | 8.16 | 16.42 | 6.82 | 3.88 |
| 3.1 | 8.18 | 5.83 | 5.18 | 5.30 | 3.72 | 4.76 | 6.28 | 7.90 | 15.44 | 6.86 | 3.45 |
| 4.15 | 10.28 | 7.53 | 6.98 | 7.34 | 5.54 | 6.90 | 8.16 | 11.28 | 17.76 | 9.22 | 4.83 |
| 6.7 | 11.84 | 7.99 | 7.68 | 8.22 | 6.36 | 7.58 | 9.54 | 12.58 | 18.82 | 10.50 | 5.28 |
| 7.5 | 12.68 | 8.59 | 7.62 | 8.82 | 6.30 | 8.52 | 9.10 | 12.02 | 19.28 | 10.36 | 6.38 |
| 7.75 | 12.88 | 9.05 | 7.96 | 9.24 | 6.76 | 8.98 | 10.04 | 12.42 | 19.19 | 11.18 | 6.88 |
| 9.6 | 13.70 | 9.13 | 9.30 | 9.62 | 7.44 | 9.40 | 11.08 | 13.58 | 20.00 | 11.88 | 7.08 |

**Table 2.** *Cont.*

| $P \cdot 10^4$ N on a Wheel | Mean Values $z \cdot 10^{-4}$ m | | | | | | | | | | |
|---|---|---|---|---|---|---|---|---|---|---|---|
| | p.p.1 | p.p.2 | p.p.3 | p.p.4 | p.p.5 | p.p.6 | p.p.7 | p.p.8 | p.p.9 | p.p.10 | p.p.11 |
| 9.85 | 13.78 | 9.35 | 9.18 | 10.32 | 7.44 | 9.88 | 11.30 | 13.82 | 20.72 | 12.04 | 6.80 |
| 10.4 | 14.33 | 10.33 | 9.28 | 10.26 | 7.76 | 10.50 | 11.32 | 14.10 | 20.74 | 12.24 | 7.65 |
| 10.55 | 14.29 | 9.75 | 9.86 | 10.50 | 8.22 | 10.04 | 11.60 | 14.78 | 20.68 | 12.92 | 7.33 |
| Standard Deviation $z \cdot 10^{-4}$ m | | | | | | | | | | | |
| 3.1 | 0.71 | 1.15 | 0.23 | 0.92 | 0.30 | 0.35 | 0.27 | 0.62 | 0.40 | 0.20 | 0.64 |
| 3.1 | 0.77 | 0.92 | 0.49 | 1.27 | 0.29 | 0.76 | 0.56 | 0.31 | 0.31 | 0.22 | 0.53 |
| 4.15 | 0.61 | 0.62 | 0.08 | 1.19 | 0.28 | 0.80 | 0.45 | 0.54 | 0.73 | 0.29 | 0.39 |
| 6.7 | 0.46 | 1.10 | 0.65 | 1.21 | 0.30 | 0.70 | 0.30 | 0.67 | 0.56 | 0.54 | 0.35 |
| 7.5 | 0.58 | 0.78 | 0.72 | 1.19 | 0.25 | 0.68 | 0.28 | 0.61 | 0.69 | 0.55 | 0.51 |
| 7.75 | 0.99 | 1.33 | 0.62 | 1.69 | 0.29 | 0.73 | 0.68 | 0.51 | 0.59 | 0.46 | 1.43 |
| 9.6 | 0.77 | 1.00 | 0.48 | 0.78 | 0.21 | 0.39 | 0.41 | 0.45 | 0.88 | 0.38 | 0.43 |
| 9.85 | 0.72 | 1.43 | 0.44 | 1.46 | 0.30 | 0.66 | 0.55 | 0.59 | 0.55 | 0.36 | 0.89 |
| 10.4 | 0.63 | 1.00 | 0.41 | 1.30 | 0.43 | 0.82 | 0.51 | 0.77 | 0.99 | 0.72 | 0.65 |
| 10.55 | 0.86 | 1.12 | 0.29 | 1.32 | 0.44 | 0.37 | 0.42 | 0.39 | 0.56 | 0.38 | 1.42 |

The magnitudes of the standard deviations do not exceed 12% of the mean value. The correlation coefficient between force and deflection was determined for the individual measurement points. This coefficient is given in Table 3.

**Table 3.** Correlation coefficient between force and deflection.

| p.p.1 | p.p.2 | p.p.3 | p.p.4 | p.p.5 | p.p.6 | p.p.7 | p.p.8 | p.p.9 | p.p.10 | p.p.11 |
|---|---|---|---|---|---|---|---|---|---|---|
| 0.99 | 0.94 | 0.95 | 0.98 | 0.97 | 0.97 | 0.97 | 0.94 | 0.98 | 0.97 | 0.93 |

The value of this coefficient allows the function P (z) to be approximated by a straight line. The approximation does not take into account the results of the $3.1 \cdot 10^4$ N load, because, as can be seen from Figure 4, the deflection of the rail from axis 5II is influenced by the load from axis 5II and the load from the axle of the coal carriage. In addition, since the stiffness characteristics of the track with subgrade are linear, there is a danger of being at the lower end of the characteristics with this load, where the stiffness is less than that determined. The average value of the resulting stiffness coefficient of the pavement with subgrade per rail is $1.54 \cdot 10^8$ N/m, and the coefficient of variation is 15%. The resulting approximation results are shown in Figure 4.

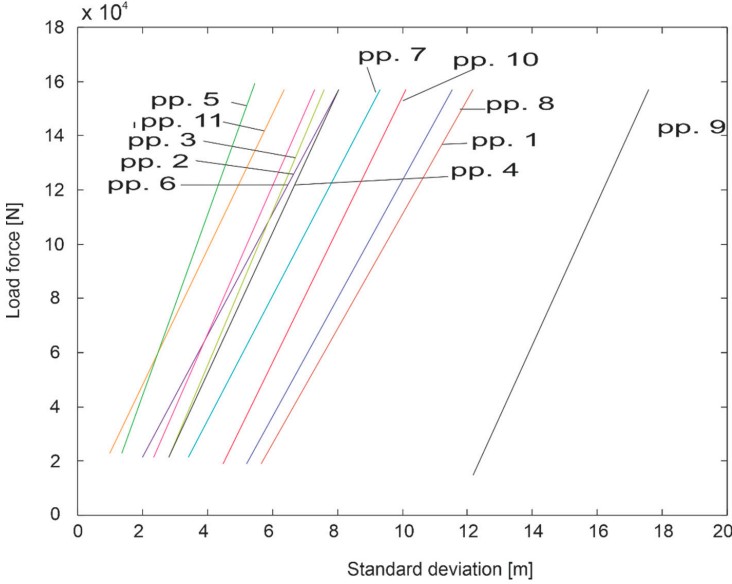

**Figure 4.** Approximation of the function P(z) by a straight line.

For a track without a turnout, the second part of the measurements was carried out on a test stand at the Railway Institute in Warsaw. This experiment was conducted to determine stiffness and damping. The damping was assumed to be dry friction.

The experiment consisted of measuring the displacement of the rail head when it was loaded and unloaded relative to the position when it was not loaded.

Loads and unloads were carried out in two variants:

– variant simulating the vertical forces acting between the railhead and the empty wagon (nominal force $2.5 \cdot 10^4$ (N)),
– variant simulating the vertical forces acting between the head of a rail and a loaded wagon (nominal force $10 \cdot 10^4$ (N)).

The head displacement was measured under static conditions in $0.5 \cdot 10^4$ N increments. In both variants, the linearity of the relationship between force and displacement under loading and unloading of the head was checked by determining the correlation coefficient—Table 4. Large values of the correlation coefficient justify the assumption of linear relationships. Straight lines were found to best describe the force-displacement relationships. The results relating to the left rail are shown in Figure 5.

**Table 4.** Correlation coefficients.

| Correlation Coefficients between: | $P_i$ a $z_l$ under Load | $P_i$ a $z_p$ under Load | $P_i$ a $z_l$ under Load | $P_i$ a $z_p$ under Load | $P_i$ a $z_l$ Medium | $P_i$ a $z_p$ Medium |
|---|---|---|---|---|---|---|
| when simulating a lighter wagon | 0.997 | 0.996 | 0.975 | 0.988 | 0.993 | 0.991 |
| when simulating a loaded wagon | 0.998 | 0.997 | 0.992 | 0.988 | 0.997 | 0.996 |

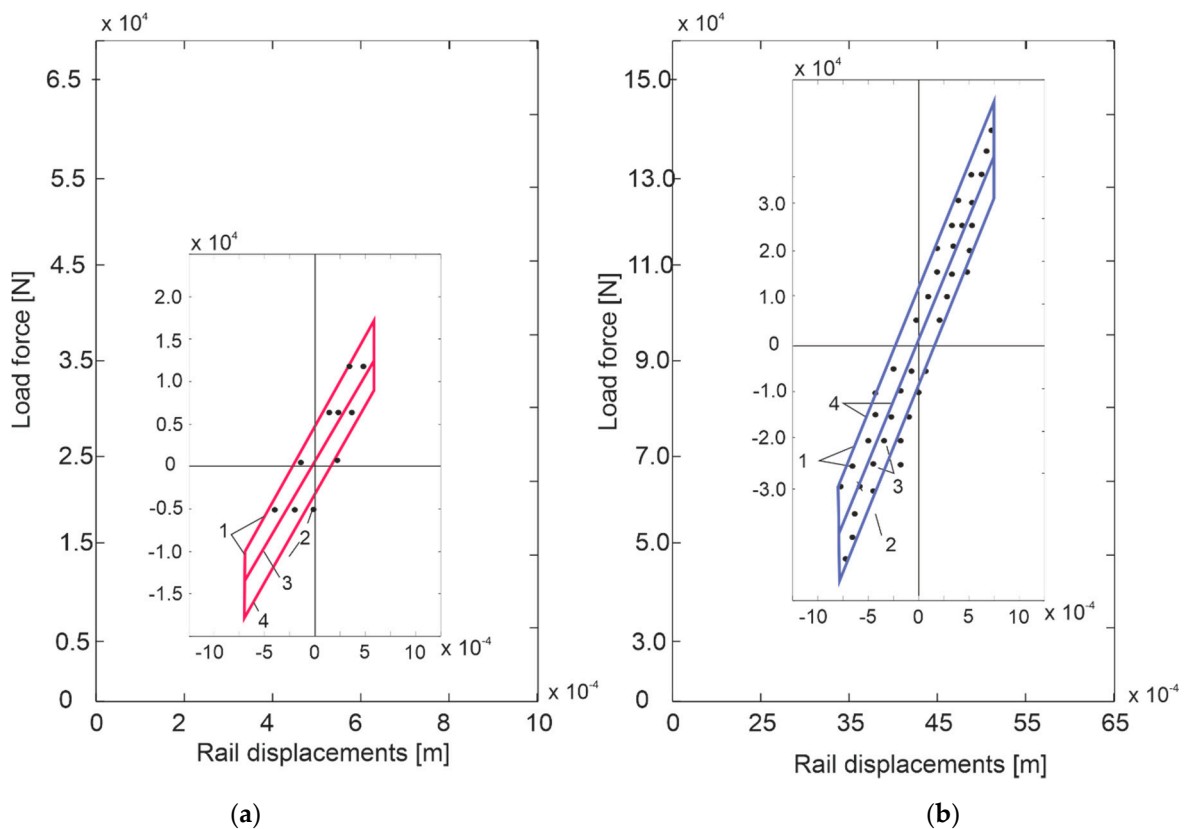

**Figure 5.** Results of checking the linear relationship between force and displacement under load and headstock relief: (**a**) variant I, (**b**) variant II; 1—load, 2—unload, 3—average of load and unload, 4—simple approximation.

The hysteresis loop of the left and right rails was then determined, with the point determined by the nominal load and the mean deflection of the empty wagon taken as the common point of the loop. These results are shown in Figure 6.

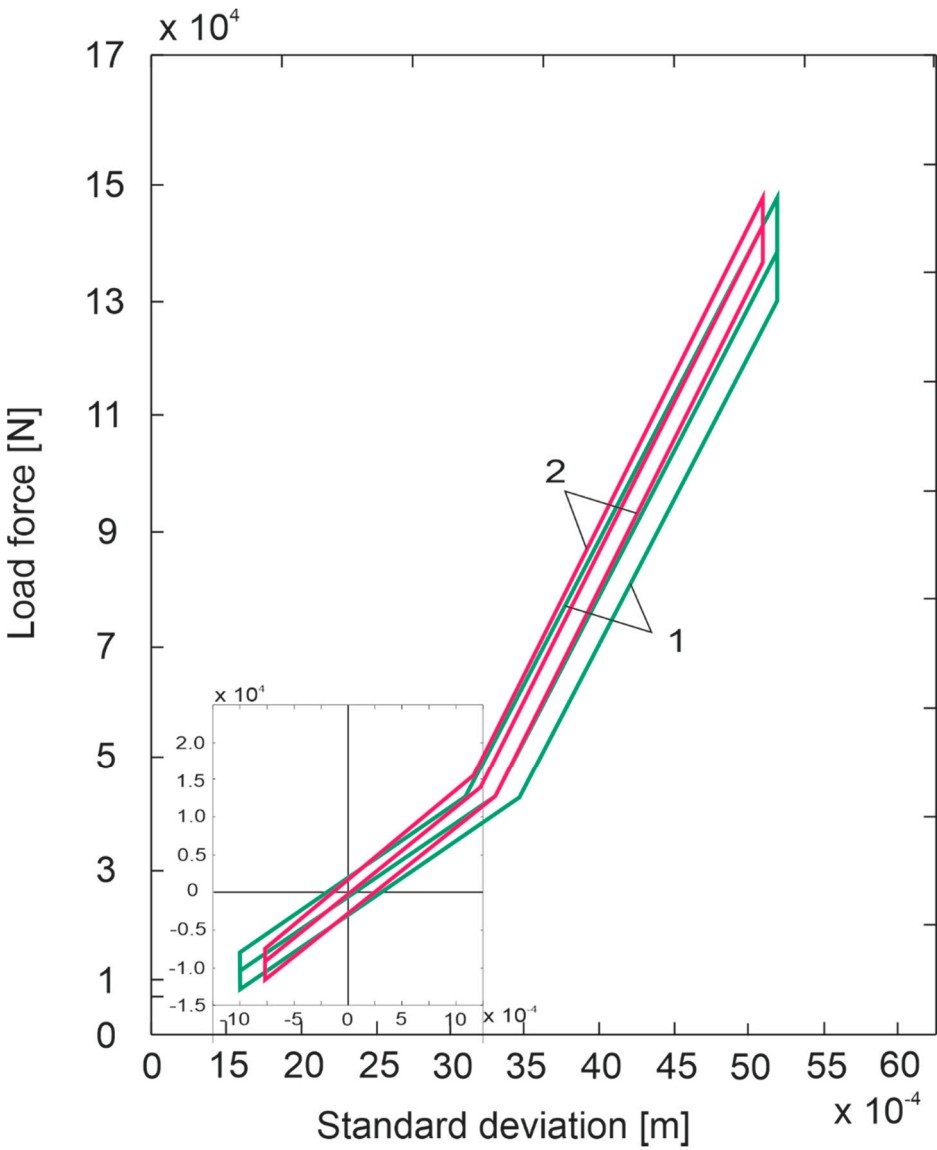

**Figure 6.** Summary of the hysteresis loops of the left (1) and right (2) rails.

On the basis of the experimental results, the following analytical description of the track forces on the wheel was considered to be the most appropriate:

$$P(z) = k_T z_T + h_T \mathrm{sgn} \dot{z}_T, \tag{1}$$

where $k_T$ and $h_T$ are constant coefficients that are different for the empty and loaded wagon (Table 5).

**Table 5.** Coefficients $k_T$ and $h_T$.

| Data Per Wheel | Empty Wagon (Nominal Load) $2.5 \cdot 10^4 \pm 1.6 \cdot 10^4$ N/on a Wheel | | Payload Wagon (Nominal Load) $10 \cdot 10^4 \pm 5.9 \cdot 10^4$ N/on a Wheel | |
|---|---|---|---|---|
| | $k_T$ N/m | $h_T$ N | $k_T$ N/m | $h_T$ N |
| Left rail | $1.8 \cdot 10^7$ | $3.6 \cdot 10^3$ | $5.1 \cdot 10^7$ | $8.6 \cdot 10^3$ |
| Right rail | $2.2 \cdot 10^7$ | $2.8 \cdot 10^3$ | $5.2 \cdot 10^7$ | $5.9 \cdot 10^3$ |
| Average values | $2.0 \cdot 10^7$ | $3.2 \cdot 10^3$ | $5.1 \cdot 10^7$ | $7.3 \cdot 10^3$ |

The results obtained for the track stiffness $k_T$ are $1.5 \cdot 10^8$ N/m. The determined damping coefficients $h_T$ vary from 3.2 to 7.3 Ns/m depending on the load. The experimental results indicate that the damping in the track is of a dry friction nature. Such an element must be kept in mind when assuming that we are dealing with linear viscous damping.

### 2.2. Stiffness Calculations in the Turnout

The second part of the study concerned the determination of stiffness parameters and moments of inertia for the turnout. This task was carried out in two ways. In the first, calculations were made on the basis of analytical relationships. In the second part, the quantities were tested on the actual object. In order to determine the stiffness values analytically, it was necessary to determine the moment of inertia of the turnout in the area of the crossing. The moment of inertia of the rail is used to describe the rotational movement of the rail vehicle-track system around the longitudinal axis of the track, usually denoted as OX.

The course of the cross-rail moment of inertia variation was determined. This waveform is shown in Figure 7.

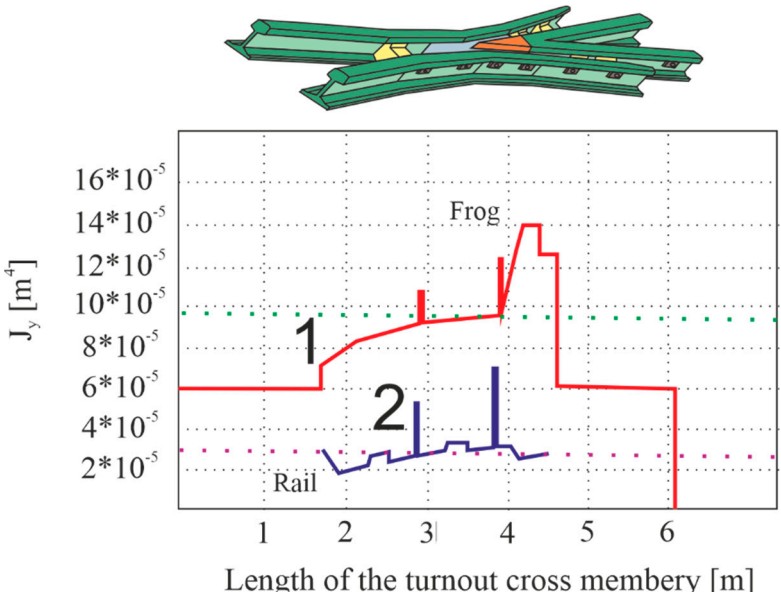

**Figure 7.** Course of variation of the moment of inertia of a turnout cross member: 1—crossbuck as a monolithic unit $\bar{J}_Y = 9.6 \cdot 10^{-5}$m$^4$, 2—crossbuck bow as an element independent of the wing rails $J_y = 3.03 \cdot 10^{-5}$m$^4$ [15].

The moment of inertia is necessary when determining the stiffness along the turnout.

Comparative calculations of the static deflections of a cross-beam loaded by a two-axle bogie with a load of 20 kN/axle were carried out, assuming that:

(a)    the cross-beam is a beam of real length with variable moment of inertia and cross-section (Figure 7 curve 1) supported on elastic supports (turnout sleepers),

(b)   the cross-beam is a beam of infinite length supported on a continuous elastic support; the moment of inertia of the cross-section can be taken as constant (averaged).

Keeping the parameters the same, it turned out that at the points of force application the difference in deflection results obtained by the two methods does not exceed 0.5%, at the other points of the cross-beam the difference is not greater than 3.5%.

With the above assumptions, the deflection of the cross member is:

$$z = \frac{k}{2U} \sum_{i=1}^{n} P_i \eta_i,$$ (2)

where

$$k = \sqrt[4]{\frac{U}{4EJ}}, \quad U = C \frac{\alpha lb}{2a}$$ (3)

$E = 2.1 \times 10^5 \ MN/m^2$—Young's modulus of the rail steel,
$J = 10 \times 10^{-5} \ m^4$—average moment of inertia of the cross section,
$P_i$—load from the next wheel,
$\eta_i$—ordinates of the influence line of settlement of an infinitely long beam resting on a continuous elastic foundation,
$U$—support factor of the rail,
$C$—ground coefficient,
$\alpha = 0.8$—ratio of the average settlement of the subgrade to its settlement under the rail,
$l = 2 \ m$—half of the average length of the turnout sleeper,
$b = 0.26 \ m$—width of the subgrade base,
$a = 0.6 \ m$—axial spacing of the turnout.

When loaded with a single force P, the deflection of the rail is:

$$z = \frac{k}{2U} P,$$ (4)

The vertical stiffness coefficient of the rail and subgrade is thus:

$$k_{tz} = \frac{P}{z} = \frac{2U}{\sqrt[4]{\frac{U}{4EJ}}},$$ (5)

As can be seen, the dependence of the $k_{tz}$ coefficient on the rail support coefficient U or the substrate coefficient C is ground coefficient.

Calculations were performed for two ground coefficients equal to $C = 50 \ MN/m^3$ and $C = 200 \ MN/m^3$. These results are shown in Figure 8.

After calculating the deflection of the cross member loaded by a single concentrated force P, the following was obtained: $k_{tz} = 50.8 \ MN/m$, he analogous value for S-60 rail is 27.6 MN/m and so the stiffness of the cross-rail is about 80% greater than that of a single rail assuming $C = 50 \ MN/m3$.

The influence of the length of the turnout gear on the value of the $k_{tz}$ coefficient is significant. The length varies from 2.6 m to 4.8 m (Figure 8), which, with a normal rail, corresponds to an increase in the $k_{tz}$ value with the adopted figures from 28 MN/m to 44 MN/m. The change in the length of the turnout sleepers resulted in a 1.58-fold increase in the stiffness of the rail along the length of the turnout (without taking into account the increase in stiffness in the region of the crossing).

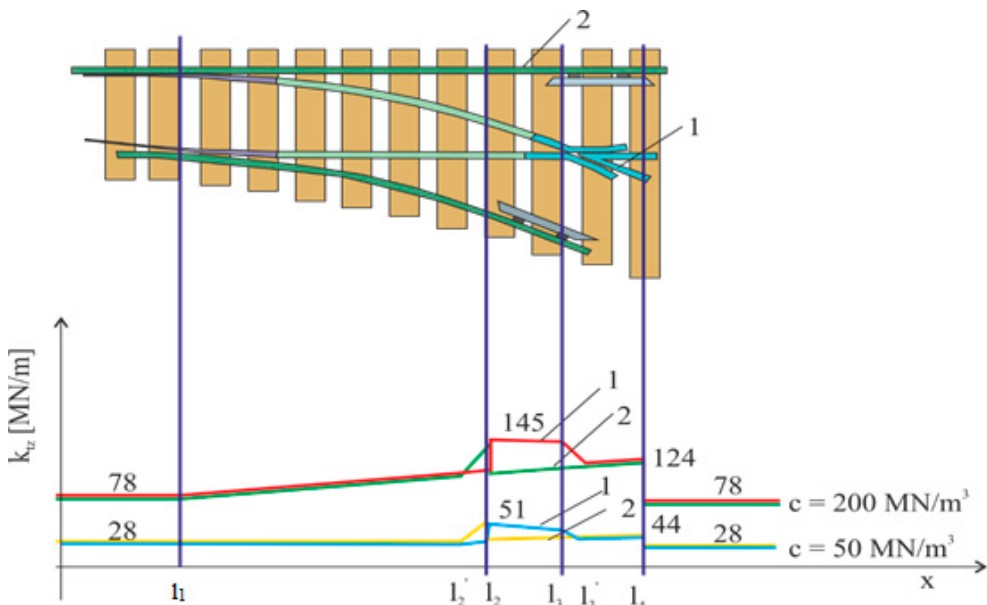

**Figure 8.** Course of variation of the vertical stiffness coefficient of the rail tracks in the turnout at different values of the ground coefficient: 1—inner track (with cross member), 2—outer track [15,16].

Figure 8 shows the variation of the turnout rail stiffness coefficient $k_{tz}$ for two values of the C-factor, assuming the following simplifications:

- the $k_{tz}$ coefficient between the specified extreme values varies linearly,
- the influence of the crossover has not been taken into account (some increase in stiffness in the section where the switch is adjacent to the bearer),
- in the crossover section, the value of $k_{tz}$ is assumed to be constant (averaged),
- the influence of the rib washers was not taken into account.

In Figure 8, the yellow line shows the course of the variation of the $k_{tz}$ coefficient, assuming that it does not vary stepwise when entering the cross-rail, but varies linearly in the section where the two rails of the cross-rail rest on common sleepers. The yellow and blue lines refer to the outer course of the rail (without the cross-rail). The lengths of the individual sections in the various S-60 ordinary turnouts are shown in Table 6.

**Table 6.** Lengths of individual sections in the various S-60 ordinary turnouts.

| Turnout Type | Length of Section [m] | | | | |
|---|---|---|---|---|---|
| | $l_1 - l'_2$ | $l'_2 - l_2$ | $l_2 - l_3$ | $l_3 - l'_3$ | $l'_3 - l_4$ |
| $R_z$ S60-190-1: 9 | 19.6 | 1.8 | 2.5 | 1.3 | 4.3 |
| $R_z$ S60-300-1: 9 | 25.6 | 1.9 | 3 | 1.4 | 5 |
| $R_z$ S60-500-1: 12 | 33.3 | 1.7 | 3.5 | 1.5 | 5.9 |
| $R_z$ S60-1200-1: 18.5 | 50.2 | 4.2 | 4.9 | 4 | 8.3 |

The results presented can be used as a basis for determining the $k_{tzl}(x)$ and $k_{tzp}(x)$ functions, which is necessary for the analysis of the parametric vibration model.

### 2.3. Experimental Determination of the Vertical Stiffness Coefficient of the Crossroads

Turnout tests were carried out at the Idzikowice station located on the Central Railway Line. A train was used at this station, consisting of a locomotive and wagons as shown in Figure 1 and Table 1. Displacements were measured using strain gauge transducers with a linear range of plus or minus 15 mm. The transducers were placed on elements that eliminated the effect of subgrade movement on the transducer. These tests were carried out to verify the analytical calculations.

The stiffness measurements used the measurement train shown in Figure 1. Loads per wheel were taken as shown in Table 1.

Deflections of three points of the cross member were recorded. The deflections were induced by the very slow passage of the measuring train.

As the distance between the individual wagon axles was considerable, so the influence of neighbouring axles on the deflection of a given point was neglected. Measurements were also made of deflections of the track on the track under the influence of a passing train for which V = 5 km/h. Displacement sensors were placed on pegs driven 1 m deep into the subgrade to reduce the influence of ballast vibrations on the sensor readings. Displacements were recorded on a magnetic recorder.

The average values of the cross-rail stiffness coefficients are given in Figure 9 and Table 7.

The average stiffness value of the plain track rail was 76 MN/m.

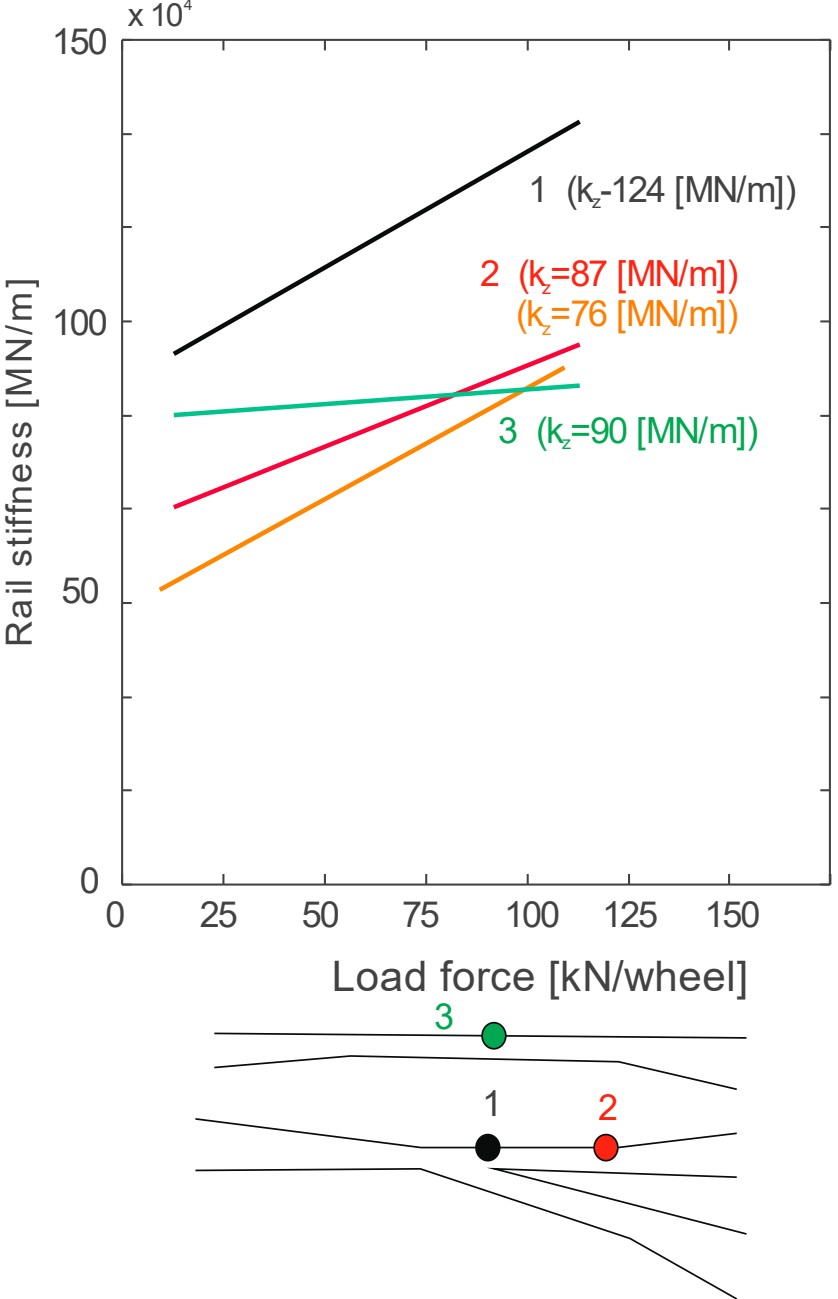

**Figure 9.** Vertical stiffness of selected points of crossbuck and plain track.

**Table 7.** Mean values of the stiffness coefficient of the crossbuck.

| Point | Location | $k_{tx}$ MN/m |
|---|---|---|
| 1 | at the blade | 124 |
| 2 | 1.3 m behind the blade | 87 |
| 3 | outer rail opposite the blade | 63 |

The experimental results presented here coincide with the results of calculations with an assumed ground coefficient of 200 MN/m$^3$ (as shown in Figure 8). Of course, the differences are small, as it was impossible to determine what the ground coefficient was in a given turnout. In paper [10], the vertical stiffness of the crossing was determined to be 70 MN/m, while the vertical stiffness of the normal rail was assumed to be between 12 and 90 MN/m. No other study in this field was found in the available bibliography.

## 3. Conclusions

The methods presented for determining the stiffness of straight track and turnout enable these parameters to be determined. The methods presented can be applied to different turnouts and different track layouts. The results obtained provide the basis for their introduction into the process of modelling the dynamics of the rail vehicle-track system. They are values of the actual stiffnesses that can occur in the track and the turnout. The vertical stiffness in the turnout in the crossover area increases by leaps and such a character should be taken into account in the process of studying the dynamics of rail vehicles passing through the turnout, especially for high-speed traffic. These quantities can provide data for the study of parametric equations, in which the stiffness is a function of the road and the dynamic process will be a function of time.

An analytical method for determining the vertical stiffness in a turnout for different ground coefficients is also presented. The method can be applied to different real-world conditions for which the ground coefficient will be determined. It also outlines how to identify pavement parameters: stiffness and damping for specific train speeds. The possibilities of using the measurement results to assess the condition of the pavement, including its vibration isolation parameters, are also shown, with examples. In general, it can be concluded that the proposed method, after the development of both the measurement technology as well as the processing of the results, offers great opportunities for practical and scientific applications and scientific applications. Namely, the processing of results and model-based identification of parameters can be performed automatically, through a counting module counting module added to the measurement modules. Furthermore, it is also possible to supplement the measuring system can also be supplemented with a set of trackside acceleration sensors, with which the "added damping" function can be determine the "added damping" function.

**Author Contributions:** Conceptualization, J.K. and R.K.; methodology, J.K. and R.K.; software, R.K.; validation, J.K.; formal analysis, J.K. and R.K.; investigation, J.K. and R.K.; resources, J.K. and R.K.; data curation, J.K. and R.K.; writing—original draft preparation, J.K. and R.K.; writing—review and editing, J.K. and R.K.; visualization, J.K. and R.K.; supervision, J.K. and R.K.; project administration, J.K. and R.K.; funding acquisition, J.K. and R.K. All authors have read and agreed to the published version of the manuscript.

**Funding:** This research received no external funding.

**Institutional Review Board Statement:** Not applicable.

**Informed Consent Statement:** Not applicable.

**Conflicts of Interest:** The authors declare no conflict of interest. The results presented were taken from the report of the Central Programme of Basic Research in Transport 0219—headed by Jerzy Kisilowski.

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
