# Peer review of "Method for Determining the Susceptibility of the Track"

_applsci, doi:10.3390/app122412534_

Round 1

Reviewer 1 Report

The paper deals with determination of the stiffness of track and turnouts. Many Some experimental tests were carried to obtain these parameters.

The paper is good, but a minor review is necessary, as follows:

Abstract

The abstract is not so clear, please rewrite.

Section 1

The novelty of this work needs to be well presented

Section 2

Which kind of special load measuring device was used?Please, give details.

How the  coefficients kt and ht were obtained?

Improve the quality of the Table 6.

It is importante include figures of the experimental tests.

Author Response

We would like to thank the reviewers for all the substantive comments that were made on the article and improved its scientific quality. Responses to the comments are included in the word file.

Reviewer 2 Report

Comments:

  1. The title “Method for the susceptibility of the track” would be better to improve like “Method for determining the susceptibility of the track” or better “susceptibility of the track for structural health estimation…”
  2. The Abstract includes many times “…were/was determined..” and so on, but there is no result or aim stated.
  3. Please improve English language and use standard terms.
  4. Line 33-34 “There were eleven such points. There were six types of wheelset loads.” – Is it literature review or the present study?
  5. Lines 42-43 “It means the degree of adaptation of the analyzed railway line section to a change operating speed.” Is it a definition of the susceptibility? The term was not used later in the paper.
  6. The substantiation of the research aims is missing at the end of the introduction.
  7. What were the measurement instrumentation in the measurement points? Was it an absolute measurement or relative to the sleepers? Please describe it in the first sections.
  8. Why Figure 2 (Turnout) is in the section2.1. Vertical stiffness measurements of straight track”?
  9. What is the aim of the Table 3. Correlation coefficient between force and deflection? It is should not be proved that the rail is bending under the wheel.
  10. Instead the “Table 2. Rail deflections” please present the displacements for the five runs.
  11. Line 96. – The function P (z) is unexpectedly introduced later in the Formula (1).
  12. What is the conclusion from the Table 4. Correlation coefficients?
  13. Fig. 5, x-axis title is missing, variant I, b) variant II; should be explicitly defined before
  14. Line 139. – “The results obtained for the track stiffness kT are 1,5∙10^8 N/m.” but the results in the Table 5 are in range 1.8-5.1e7 N/m.
  15. Figure 7 is overdimensionned. The term “turnout cross membery” is not clear – please improve the language
  16. Line 178 and 179 should be added after the formulas not before.
  17. Compare Line 180 – “C - ground coefficient”, Line 191 “substrate coefficient C”. Please show the formulas fpr its calcilation
  18. Table 6 is unreadable.
  19. What is the parametric model mentioned in Line 223?
  20. Formula (6) is not a formula. Line 198 “After calculating the deflection..” where is the deflection calculated?
  21. In the Fig.9 the rail stiffness is variable while in the table 5 is constant.
  22. The conclusions are not supported with the main part of the paper. The aim and the results are unclear. What is the improvement of the maintenance of monitoring?
  23. Line 317 “B. E. Gluberg, Badanie zestawów koÅ‚owych na rozjazdach kolejowych, WNIIST, 2, 1997” – The author’s name is Glusberg B. E. – please add the English reference

Author Response

(The authors gave the same response as above.)

Round 2

Reviewer 2 Report

The manuscript is substantially improved. However, there are still shortcomings that could be improved. The literature review and especially the reference list with 16 items (and a half of them own) are very short. It is recommended to improve the literature review with new relevant sources that are also presented in MDPI journals (Identification of sleeper support conditions…, Mechanism of sleeper dynamic impact in zones with unsupported sleepers…, Optical Rail Surface Crack Detection…).

Author Response

The manuscript is substantially improved. However, there are still shortcomings that could be improved. The literature review and especially the reference list with 16 items (and a half of them own) are very short. It is recommended to improve the literature review with new relevant sources that are also presented in MDPI journals (Identification of sleeper support conditions…, Mechanism of sleeper dynamic impact in zones with unsupported sleepers…, Optical Rail Surface Crack Detection…).

Answer:

Thank you for your fair comment, all articles have been quoted.

The paper [17] presents a method for identifying voids under the sleeper, i.e. areas where the stiffness of the track is reduced, and these areas are treated as damage to the track. This paper also presents a comprehensive analysis of this type of subject in a number of different works. None of these works deal with the determination of track stiffness, especially in the area of turnouts. Therefore, the authors do not deal with a detailed analysis of this topic. The paper [18] presents issues concerning dynamic analyses of track with hollows under the sleeper. This paper addresses the study of dynamic phenomena in the rail vehicle-track contact for sleeper voids. The works [17] and [18] deal with the study of phenomena when there are places in the track where the sleeper is not supported by ballast. This task is an element concerning emergency conditions. The authors in this paper deal with track conditions where there are no voids. In paper [19], the authors deal with track failures. This is also a track failure condition. This type of study was not dealt with by the authors of this paper. The paper presents [20] the theoretical and experimental studies of the ballast consolidation under the vibration loading of the sleeper. The practical laboratory study is given by the 1:2.5 scaled physical model of one sleeper and the corresponding ballast layer box. The measurements of ballast pressure and deformations under the vibration loading in the ballast layer and the photogrammetric recording of the ballast flow are carried out.
